# Cannabidiol Mitigates Deoxynivalenol-Induced Intestinal Toxicity by Regulating Inflammation, Oxidative Stress, and Barrier Integrity

**DOI:** 10.3390/toxins17050241

**Published:** 2025-05-12

**Authors:** Lingchen Yang, Tristan Decas, Yuhang Zhang, Imourana Alassane-Kpembi

**Affiliations:** 1Centre de Recherche en Infectiologie Porcine et Avicole (CRIPA), Faculté de Médecine Vétérinaire, Université de Montréal, Saint-Hyacinthe, QC J2S 2M2, Canada; lcyang@hunau.edu.cn (L.Y.); tristan.decas@gmail.com (T.D.); zhangyuhang@stu.hunau.edu.cn (Y.Z.); 2College of Veterinary Medicine, Hunan Agricultural University, Changsha 410128, China; 3École Nationale Vétérinaire de Toulouse ENVT, Université de Toulouse, F-31076 Toulouse, France

**Keywords:** deoxynivalenol, cannabidiol, intestinal epithelial cells, oxidative stress, dose-dependent effects

## Abstract

The deoxynivalenol (DON) mycotoxin poses serious health risks, especially to swine, which are highly susceptible to intestinal damage. Existing strategies to counteract DON toxicity remain insufficient. This study aimed to evaluate the protective effects of cannabidiol (CBD), a phytocannabinoid with anti-inflammatory properties, against DON-induced intestinal toxicity in porcine intestinal epithelial cells. Using differentiated and proliferating porcine intestinal epithelial cells (IPEC-J2), we evaluated CBD (2.5–5 μM) against DON (0.5–50 μM) through viability assays, apoptosis markers (*Bax*/*Bcl-2* ratio), inflammatory mediators (*NFκB*, *IL-6*, *COX-2*), oxidative stress indicators (*TXNIP*, *SOD1*, *CAT*), tight junction gene expression (*Claudin-1*, *Occludin*), and barrier permeability. DON exhibited dose- and time-dependent cytotoxicity (IC_50_ = 2.60 μM at 24 h; 1.07 μM at 48 h). Pre-treatment with 5 μM CBD restored cell viability at low DON concentrations (0.5–2 μM) but failed at ≥8 μM. In differentiated cells, CBD suppressed apoptosis (reduced *Bax*/*Bcl-2* ratio), oxidative stress (downregulated *TXNIP*; restored *CAT* expression), and inflammation (decreased *IL-6* and *COX-2*) under high-dose DON (50 μM), while enhancing tight junction protein expression and barrier integrity at 5 μM DON. Conversely, in proliferating cells, CBD exacerbated apoptosis (elevated *Bax*/*Bcl-2* ratio) and inflammatory responses (upregulated *IL-6* and *COX-2*) at subtoxic levels of DON (2 μM). CBD alone induced cytotoxicity at ≥10 μM. Our findings demonstrate that CBD exhibits context-dependent efficacy, providing protection in differentiated epithelia under moderate DON exposure (≤5 μM) but exhibiting detrimental effects in proliferating cells. Its narrow therapeutic window and paradoxical actions necessitate cautious application. These findings position CBD as a potential adjunctive therapy for DON detoxification but highlight critical limitations for standalone use.

## 1. Introduction

Mycotoxins, which are secondary metabolites produced by fungi, pose a formidable threat to global crops, with over 300 identified substances, among which 20 are classified as significant food safety concerns [1]. Approximately 25% of global crops are contaminated with mycotoxins, endangering human and animal health, as well as food security [2]. Deoxynivalenol (DON) is the most prevalent *Fusarium* mycotoxin [3]. It exhibits widespread contamination in swine feed, with detection rates reaching 94% and concentrations ranging from 100 to 3000 μg/kg under field conditions [4,5]. Despite stringent regulatory measures in many countries, the DON contamination levels have shown a worrying upward trend in recent years [6].

The ingestion of DON-contaminated feed leads to a range of toxic effects in domestic animals, particularly swine, manifesting as immunotoxicity, alterations in neuroendocrinology, anorexia, malabsorption, vomiting, and weight loss, with the severity varying according to the exposure dose [7,8]. Swine are highly susceptible to DON due to its high bioavailability and rapid absorption, leading to inflammation, oxidative stress, apoptosis, and pyroptosis, even at low exposure levels [9,10,11]. DON induces toxic effects on multiple systems in swine, with the small intestine and liver being the primary target organs [12]. The intestinal tract, which is crucial as a defense barrier against exogenous contaminants [13], faces impairments in epithelial cell morphology, a reduced intestinal microbial composition and diversity, increased permeability, and disrupted barrier function due to DON exposure [14,15,16]. Consequently, the development of formulations to counteract DON’s detrimental effects becomes paramount.

Several strategies have been explored to mitigate DON toxicity, including chemical compounds such as schisandrin A [17], probiotics such as *Saccharomyces cerevisiae* boulardii [18], and advanced degradation technologies such as photocatalysis [19]. However, these methods have limitations, such as high costs, residual toxicity, nutrient loss, and a dependence on complex equipment, restricting their practical application to alleviate DON-induced damage [4]. Cannabidiol (CBD), a phytocannabinoid derived from *Cannabis sativa*, has gained attention for its anxiolytic, anti-inflammatory, anti-emetic, antipsychotic, and antiepileptic properties [20,21]. The anxiolytic and antidepressant effects of CBD may improve animal welfare by mitigating stress associated with intensive swine farming [22]. Moreover, 10 μM CBD has been demonstrated to modulate the inflammatory response in primary ruminal epithelial cells (REC) by reducing pro-inflammatory cytokine expression, showing potential as both a treatment and a prophylactic [23]. Therefore, we hypothesized that CBD could protect the porcine intestinal epithelium from DON-induced damages.

This study systematically evaluated CBD’s protective effects against DON-induced intestinal toxicity using IPEC-J2 cells, a porcine intestinal epithelial model. We determined CBD’s optimal cytoprotective concentration and investigated its modulation of apoptosis, inflammation, oxidative stress, and tight junction integrity. Our findings provide critical insights into CBD’s context-dependent benefits and limitations, informing strategies to address mycotoxin contamination in swine production.

## 2. Results

### 2.1. Low Concentrations of CBD, Unlike DON, Have No Effects on Cell Viability

DON is widely recognized for its cytotoxicity across diverse cell lines. In this study, we confirmed the impact of graded concentrations of DON on the viability of IPEC-J2 monolayers at 24 and 48 h (Figure 1A). Starting from 0.5 μM, DON dose-dependently reduced cell viability at both time points (*p* < 0.001). Furthermore, the IC_50_ values were determined as 2.60 ± 0.32 μM at 24 h and 1.07 ± 0.12 μM at 48 h. Consequently, DON concentrations of 0.5, 2, and 8 μM were selected for subsequent experiments aimed at investigating the protective effects of CBD.

In order to establish the range of non-toxic concentrations of CBD, we evaluated cell viability across graded concentrations of CBD using the MTT assay (Figure 1B). CBD concentrations starting from 10 μM significantly reduced cell viability at 24 and 48 h (*p* < 0.001), whereas 2.5 and 5 μM CBD exhibited no significant effects on cell viability (*p* > 0.05). Consequently, 2.5 and 5 μM of CBD were selected for the subsequent experiment.

### 2.2. CBD Mitigated the Cell Viability Alteration Induced by DON in IPEC-J2 Cells

To evaluate the protective effects of CBD against DON-induced cytotoxicity, IPEC-J2 cells were pre-treated with 2.5 or 5 μM CBD for 24 h before DON exposure. The resulting cell viabilities are shown in Figure 2. As expected, DON treatment alone (0.5, 2, and 8 μM) significantly reduced IPEC-J2 cell viability in a dose- and time-dependent manner at both 24 and 48 h compared to the control (*p* < 0.001) (Figure 2A,B). CBD pre-treatment exhibited a protective effect, partially restoring cell viability. At 0.5 μM DON, 2.5 μM CBD completely reversed the cytotoxicity at 24 h, bringing the cell viability back to control levels (*p* > 0.05), but only 5 μM CBD significantly improved cell viability compared to DON alone at 48 h (*p* < 0.001).

At higher DON concentrations (2 and 8 μM), CBD still conferred partial protection, but the effect was less pronounced. At 2 μM DON, both 2.5 and 5 μM CBD significantly increased cell viability at 24 h (*p* < 0.001); however, the protective effect was only observed at 5 μM CBD after 48 h (*p* < 0.001) (Figure 2B). At 8 μM DON, CBD pre-treatment partially rescued cell viability, but the cells remained significantly less viable than the control group (*p* < 0.001), suggesting that CBD’s protective effects are limited at high DON concentrations. Based on these results, 5 μM CBD demonstrated stronger protective effects and was selected for further experiments.

### 2.3. CBD Exerts Divergent Effects on Apoptotic Regulation Depending on Cellular Differentiation State

To assess the impact of CBD on DON-induced apoptosis in intestinal cells, we analyzed the *Bax*/*Bcl-2* ratio in both growing (Figure 3A) and differentiated (Figure 3B) IPEC-J2 cells, as this ratio is a key indicator of apoptotic susceptibility.

Growing IPEC-J2 cells exhibited inherent resistance to DON-induced apoptosis, as evidenced by the unchanged *Bax*/*Bcl-2* ratios across all tested DON concentrations (0.5–8 μM; *p* > 0.05 relative to control; Figure 3A). This resistance may stem from their active proliferative state and robust stress response pathways. Strikingly, CBD pre-treatment (5 μM) increased the *Bax*/*Bcl-2* ratio at 2 μM DON (*p* < 0.01) compared to the DON-alone group, suggesting a paradoxical pro-apoptotic effect under subtoxic DON exposure. This finding challenges the assumption of CBD’s universal anti-apoptotic properties and highlights its potential to exacerbate cellular stress in immature epithelial cells.

In contrast, differentiated IPEC-J2 cells showed a dose-dependent apoptotic response to DON. Compared to the control, high-dose DON (50 μM) significantly elevated the *Bax*/*Bcl-2* ratio (*p* < 0.01; Figure 3B), while lower concentrations (0.5–5 μM) had no effect. CBD pre-treatment reduced the *Bax*/*Bcl-2* ratio at all tested DON concentrations (0.5–50 μM; *p* < 0.05–0.001; Figure 3B), demonstrating anti-apoptotic efficacy exclusively in mature, differentiated epithelia. Notably, CBD pre-treatment alone (without DON exposure) significantly increased the *Bax*/*Bcl-2* ratio in differentiated cells compared to untreated controls (*p* < 0.01; Figure 3B), indicating a potential intrinsic pro-apoptotic effect under baseline conditions. These results underscore CBD’s differentiation state-dependent activity, where its protective effects emerge only in cells mimicking a physiologically mature intestinal barrier. The contrasting outcomes between growing and differentiated cells emphasize that CBD’s role in apoptosis regulation is not universally protective. This dual behavior—protective against DON-induced apoptosis yet pro-apoptotic in toxin-free conditions—suggests that CBD’s action is tightly linked to the cellular stress context. While it mitigates DON-induced apoptosis in differentiated epithelia, it may inadvertently amplify apoptotic signaling in proliferating cells, potentially compromising intestinal repair mechanisms under low-level toxin exposure. This duality necessitates careful consideration of CBD’s application in dynamic intestinal environments.

### 2.4. CBD Modulates DON-Induced Inflammatory Responses in IPEC-J2 Cells

To assess the dual role of CBD in DON-triggered inflammatory responses, we analyzed the expression of key pro-inflammatory mediators (*NFκB*, *IL-6*, *COX-2*) in growing and differentiated IPEC-J2 cells (Figure 4A,B).

In growing IPEC-J2 cells, only high-dose DON (8 μM) significantly upregulated *NFκB*, *IL-6*, and *COX-2* expression (*p* < 0.001 compared to control; Figure 4A). Lower DON concentrations (0.5–2 μM) failed to induce inflammation. Strikingly, CBD pre-treatment (5 μM) amplified *NFκB*, *IL-6*, and *COX-2* expression at 2 μM DON (*p* < 0.01 and *p* < 0.001 compared to DON alone; Figure 4A), revealing a pro-inflammatory effect under subtoxic conditions. This paradoxical response suggests that CBD may sensitize proliferating epithelial cells to inflammatory signaling at low toxin exposure levels.

In differentiated IPEC-J2 cells, DON induced a dose-dependent inflammatory response, with maximal *NFκB*, *IL-6*, and *COX-2* activation at 50 μM DON (*p* < 0.001 compared to control; Figure 4B). CBD pre-treatment suppressed *IL-6* and *COX-2* expression at this high toxin dose (*p* < 0.05 compared to DON alone; Figure 4B), demonstrating anti-inflammatory activity. However, at lower DON concentrations (0.5–5 μM), CBD potentiated *NFκB*, *IL-6*, and *COX-2* expression (*p* < 0.05–0.001 compared to DON alone; Figure 4B), mirroring its pro-inflammatory role observed in growing cells. The findings highlight CBD’s context-dependent duality in modulating intestinal inflammation. While it suppresses high-dose DON-induced inflammation in differentiated epithelia—a physiologically relevant barrier model—it exacerbates inflammatory signaling at subtoxic DON levels, irrespective of the cellular differentiation state. This biphasic behavior underscores the importance of the toxin exposure intensity and cellular maturity in determining CBD’s net immunomodulatory impact.

### 2.5. CBD Has a Limited Effect on DON-Induced Oxidative Stress

To evaluate the effects of CBD in DON-induced oxidative stress, we analyzed markers of oxidative damage (*TXNIP*) and antioxidant defense (*SOD1*, *CAT*) in growing and differentiated IPEC-J2 cells (Figure 5A,B).

In growing IPEC-J2 cells, DON triggered a dose-dependent increase in *TXNIP* expression, peaking at 8 μM (*p* < 0.001 compared to control; Figure 5A). CBD pre-treatment (5 μM) exacerbated *TXNIP* expression at low DON concentrations (0.5–2 μM; *p* < 0.05–0.001 compared to DON alone) but suppressed it at 8 μM DON (*p* < 0.001 compared to DON alone; Figure 5A), revealing a concentration-dependent duality. While DON suppressed *CAT* expression at 2–8 μM (*p* < 0.05–0.001 compared to control), CBD restored *CAT* expression at 2 μM DON (*p* < 0.001 compared to DON alone), suggesting partial antioxidant rescue under moderate stress.

In differentiated IPEC-J2 cells, DON induced the maximal *TXNIP* upregulation at 50 μM (*p* < 0.001 compared to control; Figure 5B). CBD pre-treatment reduced *TXNIP* expression at both 5 and 50 μM DON (*p* < 0.001 compared to DON alone; Figure 5B), demonstrating consistent antioxidative effects in mature epithelia. Furthermore, CBD enhanced *SOD1* expression at 0.5–5 μM DON (*p* < 0.05–0.001 compared to DON alone) and upregulated *CAT* expression across these concentrations (*p* < 0.05–0.001 compared to DON alone), reinforcing its role in boosting endogenous antioxidant defenses under physiologically relevant toxin exposure.

### 2.6. CBD Enhances Tight Junction Gene Expression Only Under Moderate DON Stress

To examine the impact of CBD on DON-induced intestinal barrier dysfunction, we analyzed the gene expression of tight junction proteins *Claudin-1*, *Claudin-4*, *Occludin*, and *ZO-1* in differentiated IPEC-J2 cells (Figure 6A–D).

DON exposure triggered the compensatory upregulation of all four tight junction proteins, peaking at 5–50 μM (*p* < 0.001 compared to control; Figure 6A–D). CBD pre-treatment amplified this response at moderate DON concentrations (1–5 μM), further elevating *Claudin-1*, *Claudin-4*, *Occludin*, and *ZO-1* expression (*p* < 0.01–0.001 compared to DON alone; Figure 6A–D). However, at the highest DON dose (50 μM), CBD failed to enhance tight junction gene expression beyond the compensatory increase induced by DON alone (*p* > 0.05), indicating the saturation of its protective capacity under severe toxin stress.

While CBD reinforces tight junction gene expression under moderate DON exposure—potentially counteracting early barrier damage—it cannot override the overwhelming disruption caused by high toxin concentrations. This suggests that CBD’s efficacy is restricted to subtoxic or early-stage DON insults.

### 2.7. CBD Preserves Barrier Integrity Against Moderate but Not Severe DON Damage

To assess CBD’s functional impact on barrier integrity, we measured the epithelial permeability over 72 h (Figure 7A–D). CBD alone did not alter the baseline permeability (Figure 7A). At 0.5 μM DON, neither DON nor CBD co-treatment affected the permeability (Figure 7B). However, at 5 μM DON, the permeability increased significantly at 72 h (*p* < 0.01 compared to control; Figure 7C). CBD co-treatment prevented this increase, maintaining the permeability at control levels (*p* > 0.05; Figure 7C), demonstrating protective efficacy under moderate stress.

In contrast, 50 μM DON caused severe permeability disruption, with a sharp rise at 72 h (*p* < 0.01 compared to control; Figure 7D). CBD co-treatment failed to mitigate this effect, as the permeability in the DON + CBD group remained elevated (*p* < 0.01 compared to control; Figure 7D), highlighting CBD’s inability to counteract high-dose DON-induced barrier failure. CBD’s barrier protection is strictly dose-dependent. It effectively stabilizes membrane integrity under moderate DON exposure but is overwhelmed by severe toxin-induced damage, underscoring its limited therapeutic scope.

## 3. Discussion

This study reveals that cannabidiol (CBD) exerts conditional and context-dependent protection against deoxynivalenol (DON)-induced intestinal toxicity, with its efficacy modulated by the toxin dose, exposure duration, and cellular differentiation state. While CBD partially mitigates DON damage under specific conditions, its benefits are counterbalanced by paradoxical effects that challenge its utility as a universal therapeutic agent.

Swine are highly susceptible to DON exposure, making IPEC-J2 cells an ideal model for the study of DON toxicity [24]. Consistent with previous findings [25,26], DON reduced IPEC-J2 viability in a dose- and time-dependent manner, with significant cytotoxicity observed at 0.5 μM—a lower threshold than previously documented. CBD pre-treatment restored viability at low DON concentrations (0.5–2 μM), but this protection was diminished at higher doses (≥8 μM) (Figure 2A,B). Notably, CBD alone induced cytotoxicity at ≥10 μM (Figure 1B), emphasizing the need for precise dosing to avoid adverse effects.

DON-induced apoptosis is primarily driven by *Bax*/*Bcl-2* dysregulation and caspase activation [27]. CBD’s modulation of apoptosis was strikingly dependent on cellular differentiation. In growing IPEC-J2 cells, CBD increased the *Bax*/*Bcl-2* ratio at 2 μM DON (Figure 3A), suggesting a pro-apoptotic effect under subtoxic stress. This contrasts with its anti-apoptotic activity in differentiated cells, where CBD reduced the *Bax*/*Bcl-2* ratio across all DON concentrations (Figure 3B). Such divergence may stem from differences in the stress response pathways between proliferating and mature epithelia. For instance, growing cells prioritize survival signaling to maintain their proliferative capacity, whereas differentiated cells activate repair mechanisms [28]. These findings caution against extrapolating CBD’s effects across heterogeneous intestinal cell populations.

In growing cells, it amplified *NFκB*, *IL-6*, and *COX-2* expression at 2 μM DON (Figure 4A), potentially exacerbating early inflammatory responses. Conversely, in differentiated cells, CBD suppressed *IL-6* and *COX-2* at 50 μM DON but enhanced inflammation at lower doses (Figure 4B). This biphasic behavior mirrors the immunomodulatory properties of cannabinoids, which can activate or inhibit *NFκB* depending on the ligand concentration and cellular context [29]. These results underscore the fact that CBD’s anti-inflammatory efficacy is contingent on the toxin intensity and epithelial maturity.

DON-induced toxicity is closely linked to oxidative stress, as evidenced by increased *TXNIP* expression and decreased *SOD1* and *CAT* levels [15]. CBD’s antioxidative effects were similarly context-dependent. In growing cells, it exacerbated *TXNIP* expression—a marker of oxidative damage—at low DON levels but suppressed it at high doses (Figure 5A). While CBD reduced *TXNIP* expression at 8 μM DON in growing cells (*p* < 0.001), the levels remained 200-fold higher than those of controls, indicating limited biological relevance. In contrast, differentiated cells exhibited consistent reductions in *TXNIP* alongside enhanced expression levels of *SOD1* and *CAT* genes (Figure 5B), suggesting that CBD stabilizes the antioxidant defenses in mature epithelia. However, even in differentiated cells, CBD’s suppression of *TXNIP* at 50 μM DON (*p* < 0.001) did not restore the baseline oxidative stress markers, underscoring its restricted efficacy under severe toxin exposure. Given previous reports that antioxidants such as selenium nanoparticles and ferulic acid mitigate DON toxicity, CBD may serve as a valuable addition to oxidative stress-targeted interventions [30,31]. However, CBD’s failure to restore *CAT* activity at ≥2 μM DON in growing cells highlights limitations under sustained oxidative stress.

CBD’s modulation of oxidative stress pathways is dependent on the cell state and toxin concentration. In growing cells, CBD paradoxically amplifies oxidative damage markers at low DON levels but mitigates them at high doses. In contrast, differentiated cells exhibit consistent antioxidant benefits, suggesting that cellular maturity critically determines CBD’s efficacy. These findings underscore the importance of the toxin exposure intensity and epithelial differentiation status in optimizing CBD-based interventions.

DON disrupts tight junction proteins, leading to increased intestinal permeability [13]. CBD reinforced tight junction gene expression (*Claudin-1*, *Occludin*) and preserved barrier integrity at 5 μM DON (Figure 6 and Figure 7C), likely by amplifying the compensatory upregulation triggered by moderate toxin exposure. However, at 50 μM DON, CBD failed to prevent permeability disruption (Figure 7D), indicating an inability to counteract severe barrier damage. Similar limitations have been reported for other barrier-stabilizing agents, where very high toxin concentrations overwhelm repair mechanisms [32].

In conclusion, CBD exhibits limited and conditional efficacy against DON-induced intestinal damage, with its benefits restricted to moderate toxin exposure in swine intestinal cells. Its paradoxical effects under subtoxic conditions and cytotoxicity at higher doses underscore the need for context-specific application. While CBD holds potential as part of a multifaceted mycotoxin mitigation strategy, its role as a standalone therapeutic remains uncertain. However, translating these findings into practical applications requires further research to optimize the CBD dosages, explore its synergistic effects with other compounds, and validate its protective effects in vivo [33]. Future studies should also investigate the molecular mechanisms of CBD within the endogenous cannabinoid system, particularly the roles of CB1 and CB2 receptors. These efforts will pave the way for the development of innovative strategies to mitigate mycotoxin-related hazards and promote animal health and welfare.

## 4. Materials and Methods

### 4.1. Chemicals and Reagents

DON was purchased from Fermentek (Jerusalem, Israel), while CBD was obtained from Sigma-Aldrich (Saint-Louis, MO, USA). Dulbecco’s Modified Eagle Medium F12 (DMEM/F12), 10% fetal bovine serum (FBS), insulin–transferrin–selenium (ITS) culture supplements (100×), penicillin–streptomycin, a 3-(4,5-dimethylthiazol-2-yl)-2,5-diphenyltetrazolium bromide (MTT) assay kit, and trypsin were obtained from Wisent Bioproducts (Saint Bruno, QC, Canada). Epidermal growth factor (EGF), dexamethasone, and dimethylsulfoxide (DMSO) were sourced from Millipore Sigma (Burlington, MA, USA).

### 4.2. Cell Culture

Upon thawing, the Intestinal Porcine Epithelial Cell Line J2 (IPEC-J2 cells) was cultured in a flask containing DMEM/F12 (10% FBS (*v*/*v*), 1× ITS culture supplements, 5 ng/mL EGF, and 1% penicillin–streptomycin (*v*/*v*)). The cells were incubated overnight at 37 °C with 5% CO_2_ in the incubator. The medium was replaced the next day, and this process continued until cellular adhesion to the flask bottom was achieved. The cell culture medium was refreshed every 2 days.

The process of cell differentiation involving IPEC-J2 cells proceeded through the following steps. Initially, the cells were cultured in DMEM/F12 medium in flasks until they attained full confluence. Subsequently, the culture medium was transitioned to a differentiated state by introducing DMEM/F12 supplemented with 1× ITS culture supplements, 5 ng/mL EGF, 1% penicillin–streptomycin, and 0.25% DEX. The medium exchange was performed every 3 days, and the induction of differentiated cells was sustained for a period of 14 days.

### 4.3. Cell Viability Assay

IPEC-J2 cells were seeded in 96-well plates (Corning Costar, Cambridge, MA, USA) at a density of 1 × 10^4^ cells in 100 μL per well. The next day, the medium was replaced, and the cells were exposed to graded concentrations of CBD (0 to 40 µM) or DON (0 to 8 µM), in FBS-free medium, for 24 or 48 h. Based on the results of this preliminary investigation, the IPEC-J2 monolayers were pre-treated with CBD (2.5 or 5 µM) for 24 h and then exposed to DON (0.5, 2, 8 µM) for 24 or 48 h.

Next, 15 µL of MTT reagent (CellTiter 96 Non-Radioactive Cell Proliferation Assay, Promega, Madison, WI, USA) was added to each well at the end of the exposure period, and the plates were maintained under standard cell culture conditions for 4 h, followed by the addition of 100 µL of STOP solution to each well. The plates were sealed and humidified overnight, and the absorbance was measured at 570 nm using the Spectra i3 plate reader from Molecular Devices (San Jose, CA, USA).

### 4.4. Phenol Red Permeability Assay

The integrity of the IPEC-2 monolayer on a transwell was evaluated using the phenol red dye test [34]. Briefly, 5 × 10^4^ cells per insert were seeded in 12-insert plates (Corning Costar, Cambridge, MA, USA), with a pore size of 3.0 µm, and allowed to differentiate for 14 days. On day 14 post-differentiation, after washing the chambers, DMEM/F12 medium with phenol red (8.10 mg/L) was added to the apical chambers containing the differentiated cells either treated with 5 µM CBD, exposed to DON (0.5 to 50 µM), or pre-treated for 24 h with 5 µM CBD and then exposed to DON (0.5 to 50 µM). All basal chambers were filled with DMEM/F12 medium without phenol red. The plates were incubated at 37 °C in a CO_2_ incubator. Subsequently, 100 μL samples of medium from the basal chambers were collected in a 96-well plate at 0, 6, 24, 48, and 72 post-exposure. Absorbance was measured at 560 nm to check for phenol red leakage through intercellular spaces, using the Spectra i3 plate reader from Molecular Devices (San Jose, CA, USA).

### 4.5. Total RNA Extraction and Real-Time qPCR (RT-qPCR)

Growing or differentiated cells IPEC-J2 cells (5 × 10^5^ cells per well) cultured in 6-well plates (Corning Costar, Cambridge, MA, USA) were either treated with 5 µM CBD or graded concentrations of DON (0.5 to 50 µM) for 24 h or pre-treated with 5µM CBD for 24 h before being exposed to the graded concentrations of DON for an additional 24 h. The culture medium was then discarded, and the cells were washed twice with PBS. Total RNA was extracted using Trizol (Bio-Rad, Hercules, CA, USA), as per the manufacturer’s guidelines. The purity and concentration of total RNA were determined using NanoDrop microvolume spectrophotometers (Thermo Fisher Scientific, Waltham, MA, USA). The RNA was reverse-transcribed into cDNA using the Wisent cDNA Synthesis Kit (Wisent, Saint-Bruno-de-Montarville, QC, Canada).

RT-qPCR was conducted using the CFX96 Real-Time System C1000 Touch Thermal Cycler from Bio-Rad (Hercules, CA, USA), employing the Power SYBR Green PCR Master Mix (Wisent, Saint-Bruno-de-Montarville, QC, Canada). Standardized thermal cycling parameters, including initial denaturation at 95 °C for 3 min, followed by 40 cycles of denaturation at 95 °C for 15 s, annealing at 59 °C for 30 s, and extension at 72 °C for 30 s, were applied for transcript amplification. Melting curve analyses were systematically conducted to confirm the product identity, and sequencing procedures further validated the novel amplicons. Experimental samples were processed in duplicate and quantified relative to the reference gene *GAPDH*, serving as the housekeeping gene. Primer details for the target genes and the housekeeping gene are provided in Table 1. Data were normalized to a calibrator sample using the Pfaffl method with correction for the amplification efficiency [35].

### 4.6. Statistical Analysis

The experiments were performed with three independent replicates, and the data are presented as the means ± standard error of the mean (SEM). Data analysis and the generation of statistical graphs were conducted using the GraphPad Prism 8 software. One-way analysis of variance (ANOVA) was used to compare the control and treated groups, followed by pairwise comparisons between the groups treated with DON and DON + CBD using Tukey’s HSD test to control for multiple comparisons. When *p* < 0.05, the difference was significant; * or #: *p* < 0.05; ** or ##: *p* < 0.01; *** or ###: *p* < 0.001.

## Figures and Tables

**Figure 1 toxins-17-00241-f001:**
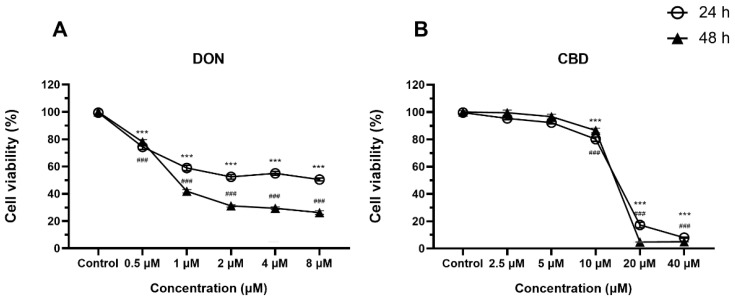
Effects of DON and CBD on IPEC-J2 cell viability at 24 and 48 h. (**A**) IPEC-J2 cells were exposed to increasing concentrations of DON (0.5–8 μM) for 24 and 48 h. Cell viability was assessed using the MTT assay and expressed as a percentage of the control group. DON significantly reduced cell viability in a dose- and time-dependent manner. (**B**) IPEC-J2 cells were treated with varying concentrations of CBD (2.5–40 μM) for 24 and 48 h. CBD alone did not significantly affect cell viability at concentrations up to 5 μM but induced cytotoxicity at higher concentrations (≥10 μM). Data are presented as mean ± SEM (*n* = 3). Statistical significance: *** *p* < 0.001 compared to the control group at 24 h; ### *p* < 0.001 compared to the control group at 48 h.

**Figure 2 toxins-17-00241-f002:**
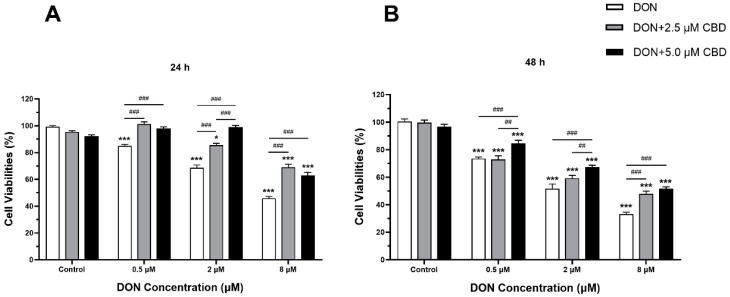
Effects of CBD on DON-induced reduction in IPEC-J2 cell viability at 24 and 48 h. (**A**) IPEC-J2 cells were pre-treated with 2.5 μM or 5.0 μM CBD for 24 h, followed by exposure to increasing concentrations of DON (0.5, 2, and 8 μM) for an additional 24 h. Cell viability was measured using the MTT assay and expressed as a percentage of the control group. (**B**) Similar experimental conditions were applied, except that cells were exposed to DON for 48 h following CBD pre-treatment. Data are presented as mean ± SEM (*n* = 3). Statistical significance: * *p* < 0.05, *** *p* < 0.001 compared to the control in the same treatment; ## *p* < 0.01, ### *p* < 0.001 between different treatments at the same DON concentration.

**Figure 3 toxins-17-00241-f003:**
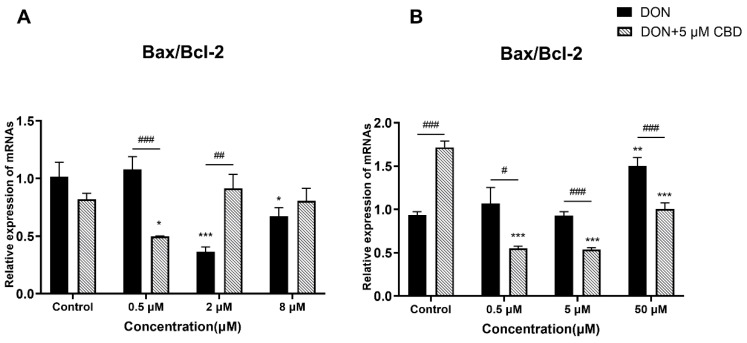
Effects of CBD on DON-induced *Bax/Bcl-2* expression ratio in growing and differentiated IPEC-J2 cells. (**A**) Relative mRNA expression of *Bax/Bcl-2* ratio in growing IPEC-J2 cells treated with DON (0.5, 2, and 8 μM) alone or pre-treated with 5 μM CBD. (**B**) Relative mRNA expression of *Bax/Bcl-2* ratio in differentiated IPEC-J2 cells treated with DON (0.5, 5, and 50 μM) alone or pre-treated with 5 μM CBD. Data are presented as mean ± SEM (*n* = 3). Statistical significance: * *p* < 0.05, *** p* < 0.01, *** *p* < 0.001 compared to the control in the same treatment; # *p* < 0.05, ## *p* < 0.01, ### *p* < 0.001 between different treatments at the same DON concentration.

**Figure 4 toxins-17-00241-f004:**
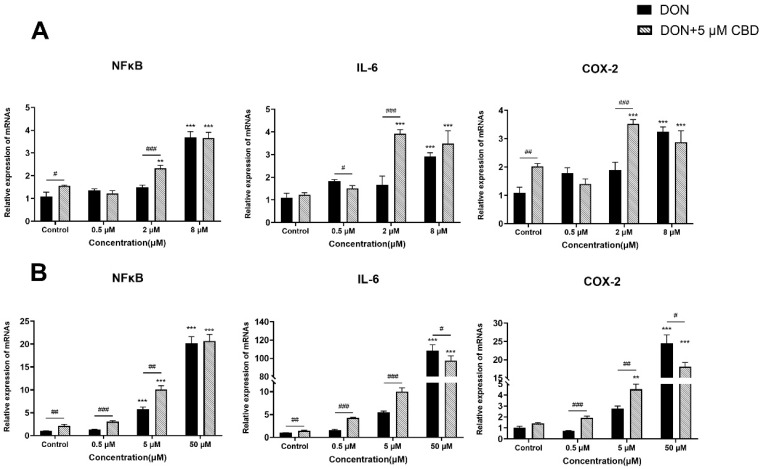
Effects of CBD on DON-induced pro-inflammatory gene expression in growing and differentiated IPEC-J2 cells. (**A**) Relative mRNA expression levels of *NFκB*, *IL-6*, and *COX-2* in growing IPEC-J2 cells treated with DON (0.5, 2, and 8 μM) alone or pre-treated with 5 μM CBD. (**B**) Relative mRNA expression levels of *NFκB*, *IL-6*, and *COX-2* in differentiated IPEC-J2 cells treated with DON (0.5, 5, and 50 μM) alone or pre-treated with 5 μM CBD. Data are presented as mean ± SEM (*n* = 3). Statistical significance: ** *p* < 0.01, *** *p* < 0.001 compared to the control in the same treatment; # *p* < 0.05, ## *p* < 0.01, ### *p* < 0.001 between different treatments at the same DON concentration.

**Figure 5 toxins-17-00241-f005:**
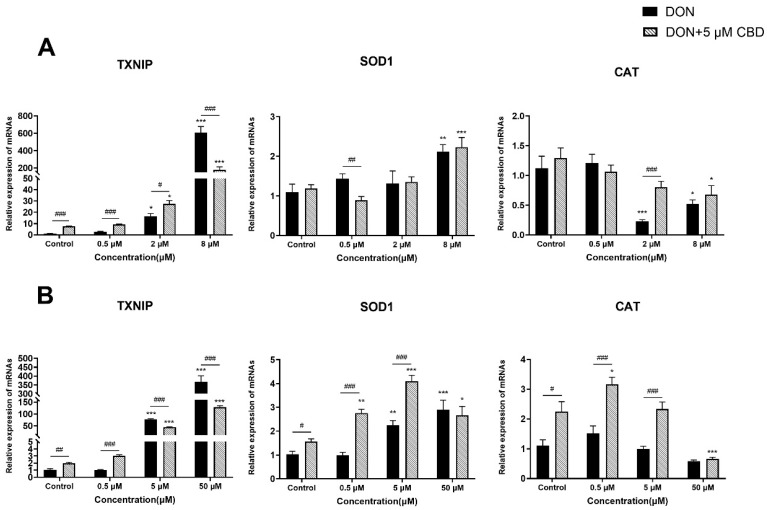
Effects of CBD on DON-induced oxidative stress-related gene expression in growing and differentiated IPEC-J2 cells. (**A**) Relative mRNA expression levels of *TXNIP*, *SOD1*, and *CAT* in growing IPEC-J2 cells treated with DON (0.5, 2, and 8 μM) alone or pre-treated with 5 μM CBD. (**B**) Relative mRNA expression levels of *TXNIP*, *SOD1*, and *CAT* in differentiated IPEC-J2 cells treated with DON (0.5, 5, and 50 μM) alone or pre-treated with 5 μM CBD. Data are presented as mean ± SEM (*n* = 3). Statistical significance: * *p* < 0.05, ** *p* < 0.01, *** *p* < 0.001 compared to the control in the same treatment; # *p* < 0.05, ## *p* < 0.01, ### *p* < 0.001 between different treatments at the same DON concentration.

**Figure 6 toxins-17-00241-f006:**
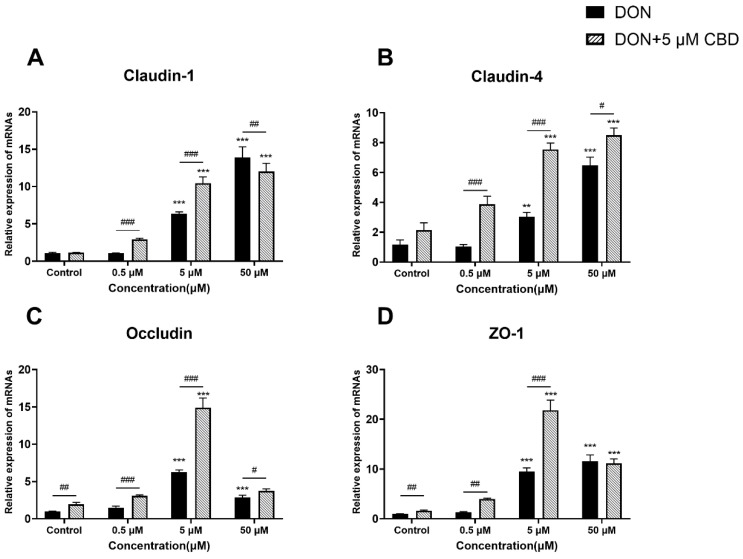
Effects of CBD on DON-induced tight junction gene expression in IPEC-J2 cells. (**A**–**D**) Relative mRNA expression levels of tight junction-related genes, including *Claudin-1* (**A**), *Claudin-4* (**B**), *Occludin* (**C**), and *ZO-1* (**D**), in IPEC-J2 cells treated with DON (0.5, 5, and 50 μM) alone or pre-treated with 5 μM CBD. Data are presented as mean ± SEM (*n* = 3). Statistical significance: ** *p* < 0.01, *** *p* < 0.001 compared to the control in the same treatment; # *p* < 0.05, ## *p* < 0.01, ### *p* < 0.001 between different treatments at the same DON concentration.

**Figure 7 toxins-17-00241-f007:**
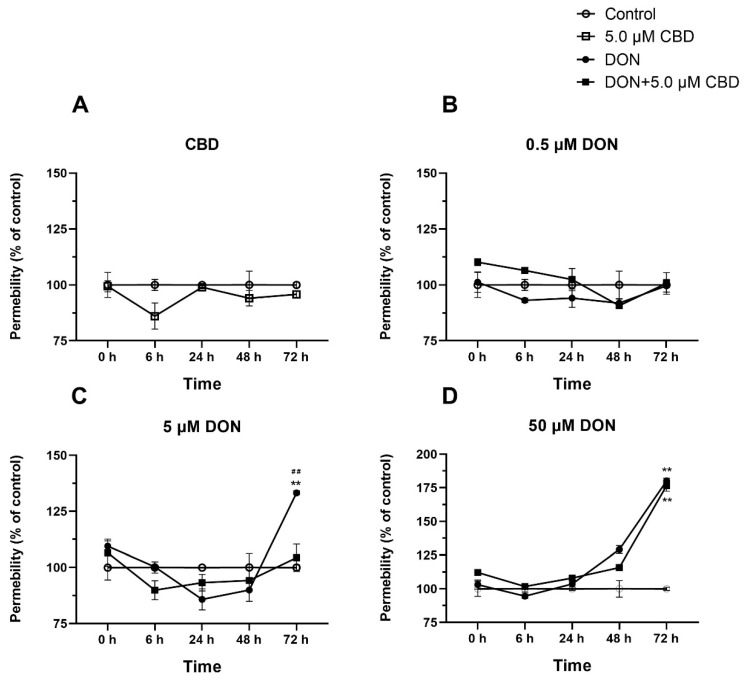
Effects of CBD on DON-induced changes in membrane permeability in IPEC-J2 cells over time. (**A**–**D**) Time-dependent changes in membrane permeability (% of control) in IPEC-J2 cells treated with 5 μM CBD alone (**A**), 0.5 μM DON alone or in combination with 5 μM CBD (**B**), 5 μM DON alone or in combination with 5 μM CBD (**C**), and 50 μM DON alone or in combination with 5 μM CBD (**D**). Data are presented as mean ± SEM (*n* = 3). Statistical significance: ** *p* < 0.01 compared to the control at the same time point; ## *p* < 0.01 compared to DON + 5 μM CBD at the same time point.

**Table 1 toxins-17-00241-t001:** Primer sequences of RT-qPCR.

Gene	Upstream Primer (5′→3′)	Downstream Primer (5′→3′)
*NF-κB*	CTCGCACAAGGAGACATGAA	ACTCAGCCGGAAGGCATTAT
*IL-6*	ACCTGCTTGATGAGAATCACC	CTTCATCCACTCGTTCTGTGA
*COX-2*	TGCGGGAACATAATAGAG	GTATCAGCCTGCTCGTCT
*TXNIP*	TTGGAGGAAAGACAGGAAAGA	AACAAAACCCCGAATCAAAG
*SOD1*	GAGACCTGGGCAATGTGAC	GAGGGAATGTTTACTGGGTGA
*CAT*	GGCTTTTGGCTACTTTGAGG	AGGGTCACGAACTGTGTCAG
*Bcl-2*	GGATAACGGAGGCTGGGATG	TTATGGCCCAGATAGGCACC
*Bax*	GCCCTTTTGCTTCAGGGTTTC	CAATGCGCTTGAGACACTCG
*Claudin-1*	TTTCCTCAATACAGGAGGGAAGC	CCCTCTCCCCACATTCGAG
*Claudin-4*	CTGCTTTGCTGCAACTGC	TCAACGGTAGCACCTTACACGTAG
*Occludin*	CTACTCGTCCAACGGGAAAG	ACGCCTCCAAGTTACCACTG
*ZO-1*	AAGCCCTAAGTTCAATCACAATCT	ATCAAACTCAGGAGGCGGC
*GAPDH*	GGGCATGAACCATGAGAAGT	TGTGGTCATGAGTCCTTCCA

## Data Availability

The original contributions presented in this study are included in this article. Further inquiries can be directed to the corresponding author.

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
