# Peer review of "Cannabidiol Mitigates Deoxynivalenol-Induced Intestinal Toxicity by Regulating Inflammation, Oxidative Stress, and Barrier Integrity"

_toxins, 2025, doi:10.3390/toxins17050241_

Round 1
Reviewer 1 Report
Comments and Suggestions for Authors
Overview:
This is an interesting study that provides some evidence that CBD might be a suitable protective agent in some circumstances against DON contamination for swine. The authors provide a logical series of experiments to support this concept. The discussion of the results is appropriately nuanced and does not overstate the significance. One area for consideration is that in some cases, a statistically significant result is achieved, but this may not be a biologically significant result. The authors would do well to consider this in appropriate places in revision.
Comments:
P1, Line 34: add “is” after the first comma; replace the second comma with a full stop
Introduction general comment. It would be useful to give some discussion of the concentrations of DON that are typically found in feeds for swine. It might also be useful to discuss the CBD concentrations that are typically used in other indications and the cost of these treatments.
P2, line 54: remove additional comma before the full stop.
P2, line 73: (p < 0.001) is repeated unnecessarily.
P2, figure 1: please check the statistics. At 0.5 uM DON and 10 and 40 uM CBD, the 24 and 48 hour samples are shown as p < 0.001 even though error bars are overlapping.
P3, figure 2 (and figures 3-5): please indicate the meaning of **, ***, ###. Also indicate the number of samples (n) and the meaning of the error bars. Each figure should stand alone without requirement to refer to other figures for these meanings.
P4, Figure 3B: a very important result that is not commented on is that CBD pre-treatment appears to significantly increase the bax/bcl-2 ratio in differentiated cells that are not exposed to DON. This is a very important observation as it suggests that there may be a clear cost to CBD treatment. Given that this is the highest increase in this ratio observed, it is odd that there is no comment on this. Also for Figure 3B, the DON concentrations used are higher than explained earlier (lines 74-5); the authors should explain why.
P5, lines 179-189. I would suggest that the authors frame the discussion in terms of both the significance of results and the effects relative to the control. For TXNIP, at 8 uM DON there is a reduced effect with CBD pre-treatment – but this is still 200 times the unexposed control and higher than any result at 2 uM DON. Therefore, although there is a statistically significant result, this likely has limited biological significance. The same is true for the differentiated cells, where at 5 and 50 uM DON there is a statistically significant CBD effect but the levels are still highly elevated.
This should be mirrored in discussion.
P9, line 276: italicise TXNIP.
P9, lines 299-300. This sentence is not justified based on the data presented. The data have been obtained from cultured cells, not the swine intestine. To make a claim of this nature, a more sophisticated model replicating the swine intestine would be needed. It would be better to say that CBD demonstrates protective effects of swine intestinal cells.
P10, line 322: when percentage concentrations are used (e.g. for FBS), indicate whether this is (v/v) or (w/v).
P10, line 341-2: explain the source of MTT and STOP solutions.
P10, line 348: provide the supplier and part number of the 12-insert plates.
P10, line 350: how much phenol red was added to the medium?
P10, line 360: add “plates” at the end of the line. Provide the supplier and part no.
Author Response
We sincerely thank the reviewer for the constructive feedback and careful review of our manuscript. Your insights have been invaluable in helping us refine the work. We have meticulously addressed each suggestion to ensure the manuscript’s accuracy, clarity, and scientific rigor, and made necessary revisions to strengthen the presentation of our research. The detailed responses to each of your comments are provided below.
Comments 1:
P1, Line 34: add “is” after the first comma; replace the second comma with a full stop
Introduction general comment. It would be useful to give some discussion of the concentrations of DON that are typically found in feeds for swine. It might also be useful to discuss the CBD concentrations that are typically used in other indications and the cost of these treatments.
P2, line 54: remove additional comma before the full stop.
P2, line 73: (p < 0.001) is repeated unnecessarily.
Response 1: We thank the reviewer for the comments. We have implemented the suggested formatting changes and included reports on DON concentrations in pig feed. As for CBD, given the limited anti-oxidative research, the author has added the following to the Introduction: "10μM CBD has demonstrated to modulate the inflammatory response in primary ruminal epithelial cells (REC) by reducing pro-inflammatory cytokine expression, showing potential as both a treatment and a prophylactic.
Comments 2:
P2, figure 1: please check the statistics. At 0.5 uM DON and 10 and 40 uM CBD, the 24 and 48 hour samples are shown as p < 0.001 even though error bars are overlapping.
Response 2: Apologies for the incorrect labeling; this has now been amended.
Comments 3:
P3, figure 2 (and figures 3-5): please indicate the meaning of **, ***, ###. Also indicate the number of samples (n) and the meaning of the error bars. Each figure should stand alone without requirement to refer to other figures for these meanings.
Response 3: We have made the required changes to the paper in line with the reviewer's comment.
Comments 4:
P4, Figure 3B: a very important result that is not commented on is that CBD pre-treatment appears to significantly increase the bax/bcl-2 ratio in differentiated cells that are not exposed to DON. This is a very important observation as it suggests that there may be a clear cost to CBD treatment. Given that this is the highest increase in this ratio observed, it is odd that there is no comment on this. Also for Figure 3B, the DON concentrations used are higher than explained earlier (lines 74-5); the authors should explain why.
Response 4: Thanks for the suggestion. The impact of CBD alone on the bax/bcl-2 ratio in differentiated IPEC-J2 cells has been described in the results. Regarding DON concentrations, different concentrations were used for growing and differentiated cells.
Comments 5:
P5, lines 179-189. I would suggest that the authors frame the discussion in terms of both the significance of results and the effects relative to the control. For TXNIP, at 8 uM DON there is a reduced effect with CBD pre-treatment – but this is still 200 times the unexposed control and higher than any result at 2 uM DON. Therefore, although there is a statistically significant result, this likely has limited biological significance. The same is true for the differentiated cells, where at 5 and 50 uM DON there is a statistically significant CBD effect but the levels are still highly elevated.
This should be mirrored in discussion.
Response 5: Appreciating the reviewer's advice, the discussion has been revised accordingly.
Comments 6:
P9, line 276: italicise TXNIP.
P9, lines 299-300. This sentence is not justified based on the data presented. The data have been obtained from cultured cells, not the swine intestine. To make a claim of this nature, a more sophisticated model replicating the swine intestine would be needed. It would be better to say that CBD demonstrates protective effects of swine intestinal cells.
P10, line 322: when percentage concentrations are used (e.g. for FBS), indicate whether this is (v/v) or (w/v).
P10, line 341-2: explain the source of MTT and STOP solutions.
P10, line 348: provide the supplier and part number of the 12-insert plates.
P10, line 350: how much phenol red was added to the medium?
P10, line 360: add “plates” at the end of the line. Provide the supplier and part no.
Response 6: We have made all the above - mentioned modifications as per the reviewer's suggestions.
Reviewer 2 Report
Comments and Suggestions for Authors
Manuscript ID: toxins-3625325 – review
Title: Cannabidiol Mitigates Deoxynivalenol-Induced Intestinal Toxicity by Regulating Inflammation, Oxidative Stress, and Barrier Integrity
In this manuscript, cannabidiol (CBD) was investigated for its potential to mitigate deoxynivalenol (DON)-induced intestinal toxicity. The topic raised is very important, because ingestion of DON-contaminated feed leads to a range of toxic effects in domestic animals, particularly swine. DON induces toxic effects on multiple systems in swine, with the small intestine and liver being the primary target organs. Cannabidiol has gained attention for its anxiolytic, anti-inflammatory, antiemetic, antipsychotic, and antiepileptic properties. It may improve animal welfare by mitigating stress associated with intensive swine farming. Therefore, the authors decided to test the mitigating effect of CBD on the impact of this mycotoxin.
The Reviewer recommends the above-mentioned paper for publication in Toxins after minor revision.
Title:
- In the Reviewer’s opinion it is better to ask questions in the title rather than provide answers, e.g. “Effect of cannabidiol on mitigating deoxynivalenol toxicity on intestinal epithelial cells by regulating inflammation, oxidative stress and barrier integrity”. However, this is only a suggestion.
Abstract:
- Lack of a clearly stated research objective. The authors have combined the aim of the study with the research methods.
Introduction:
- Something in lines 33-35 is wrong: „Deoxynivalenol (DON), the most prevalent Fusarium mycotoxin [3], Despite stringent…..”, unclear whether it was meant to be one or two sentences.
- The authors wrote that: „DON contamination levels have shown a worrying upward trend in recent years [4]”, but cited a paper from 2013. Recommended updating the reference for DON trends (2013 is outdated for "recent years").
Results:
- The description of the results is sufficient.
- Results were presented in several well-prepared figures.
- The chapter contains the most important conclusions of the research performed. At the same time, the authors summarized the obtained results of laboratory analysis in a good way.
Discussion:
- In the Reviewer's opinion, the chapter was prepared very logically.
Materials and Methods:
- According to the Reviewer, the chapter contains all the necessary information.
References:
- The authors have cited 34 literature items that align with the manuscript's content. Most of these publications are from the past 10 years. While the number of references is not extensive, the reviewer acknowledges that research on CBD application remains a relatively novel field.
Author Response
We sincerely thank the reviewer for the constructive feedback and careful review of our manuscript. Your insights have been invaluable in helping us refine the work. We have meticulously addressed each suggestion to ensure the manuscript’s accuracy, clarity, and scientific rigor, and made necessary revisions to strengthen the presentation of our research. The detailed responses to each of your comments are provided below.
Comment 1:
Title: In the Reviewer’s opinion it is better to ask questions in the title rather than provide answers, e.g. “Effect of cannabidiol on mitigating deoxynivalenol toxicity on intestinal epithelial cells by regulating inflammation, oxidative stress and barrier integrity”. However, this is only a suggestion.
Response 1: We thank the reviewer for this constructive suggestion. However, we think that keeping the title in its current form provides in few words an overview of the main findings of the study, which could be more appealing for the readership.
Comment 2:
Abstract: Lack of a clearly stated research objective. The authors have combined the aim of the study with the research methods.
Response 2: We appreciate your feedback. The abstract has been restructured to separate the research objective from the methodology.
Comment 3:
Introduction: Something in lines 33-35 is wrong: „Deoxynivalenol (DON), the most prevalent Fusarium mycotoxin [3], Despite stringent…..”, unclear whether it was meant to be one or two sentences.
The authors wrote that: „DON contamination levels have shown a worrying upward trend in recent years [4]”, but cited a paper from 2013. Recommended updating the reference for DON trends (2013 is outdated for "recent years").
Response 3: The text mentioned above has been revised. We have replaced the outdated citation [4] with recent literature supporting DON trends (Sumarah, 2022).
Comment 4:
Results:
- The description of the results is sufficient.
- Results were presented in several well-prepared figures.
- The chapter contains the most important conclusions of the research performed. At the same time, the authors summarized the obtained results of laboratory analysis in a good way.
Discussion:
- In the Reviewer's opinion, the chapter was prepared very logically.
Materials and Methods:
- According to the Reviewer, the chapter contains all the necessary information.
References:
- The authors have cited 34 literature items that align with the manuscript's content. Most of these publications are from the past 10 years. While the number of references is not extensive, the reviewer acknowledges that research on CBD application remains a relatively novel field.
Response 4: Thank you for your positive feedback.
Reference:
Sumarah, M.W., (2022). The Deoxynivalenol Challenge. J Agric Food Chem 70, 9619-9624.